# The Possible Role of Food and Diet in the Quality of Life in Patients with COPD—A State-of-the-Art Review

**DOI:** 10.3390/nu15183902

**Published:** 2023-09-07

**Authors:** Mónika Fekete, Tamás Csípő, Vince Fazekas-Pongor, Madarász Bálint, Zoltán Csizmadia, Stefano Tarantini, János Tamás Varga

**Affiliations:** 1Department of Public Health, Faculty of Medicine, Semmelweis University, 1089 Budapest, Hungary; fekete.monika@med.semmelweis-univ.hu (M.F.); csipo.tamas@med.semmelweis-univ.hu (T.C.); pongor.vince@med.semmelweis-univ.hu (V.F.-P.); balintmadarasz1993@gmail.com (M.B.); 2Faculty of Health Sciences, University of Pécs, 7621 Pécs, Hungary; penituki@gmail.com; 3Department of Neurosurgery, The University of Oklahoma Health Sciences Center, Oklahoma City, OK 73104, USA; stefano-tarantini@ouhsc.edu; 4Department of Health Promotion Sciences, College of Public Health, The University of Oklahoma Health Sciences Center, Oklahoma City, OK 73104, USA; 5Peggy and Charles Stephenson Oklahoma Cancer Center, Oklahoma City, OK 73104, USA; 6Department of Pulmonology, Semmelweis University, 1083 Budapest, Hungary

**Keywords:** chronic obstructive pulmonary disease, nutritional intervention, protein, omega-3 polyunsaturated fatty acids, randomized controlled trial

## Abstract

Diet has been described as a modifiable risk factor for the development and progression of chronic diseases, and emerging evidence increasingly points to its preventive and therapeutic role in chronic obstructive pulmonary disease (COPD). While the relationship between the underlying disease and diet is natural in conditions such as metabolic disorders, obesity, diabetes, etc., the direct effect is not so evident in chronic obstructive pulmonary disease. Poor diet quality and the development of nutrient deficiencies in respiratory diseases, including COPD, can be associated with disease-specific factors such as the exacerbation of respiratory symptoms. These symptoms can be improved by dietary interventions, leading to positive changes in the pathogenesis of the disease and the quality of life of patients. Therefore, our aim was to review the latest randomized controlled trials (RCTs) of dietary interventions in chronic respiratory patients and describe their effects on respiratory function, physical activity, systemic inflammatory parameters, and quality of life. We conducted a literature search on dietary interventions for COPD patients in the PubMed, ClinicalTrials.gov, and Cochrane Central Register of Controlled Trials (CENTRAL) databases, focusing on publications from 1 July 2018 to 1 July 2023. We used specific keywords and MESH terms, focusing on RCTs. A total of 26 articles and 1811 COPD patients were included in this review. On the basis of our findings, dietary interventions, in particular components of the Mediterranean diet such as protein, omega-3 polyunsaturated fatty acids, and vegetables, appear to have beneficial effects in patients with chronic respiratory diseases, and their application is beneficial. However, long-term follow-up studies are still needed to examine the effects of dietary interventions in this patient population.

## 1. Introduction

Chronic obstructive pulmonary disease (COPD), characterized by persistent and generally progressive bronchial obstruction, is a preventable and treatable public health issue [1]. COPD is a major chronic disease: its prevalence is widespread with high mortality, and its long-term treatment is extremely expensive [1,2]. By 2020, it became the third leading cause of death globally [2]. Severe COPD develops in 15–20% of chronic smokers, whereas the majority of smokers experience mild impairment of lung function [3]. Current pharmacotherapy provides symptomatic relief, but has limited impact on COPD progression [4]. The management of COPD involves a team of specialists, including a pulmonologist, a respiratory rehabilitation expert, a general practitioner, a physiotherapist, a psychologist, as well as a dietitian. Patients with COPD may require a special diet based on their nutritional status or may need dietary adjustments in terms of nutrient composition and/or consistency [5,6]. The initiation of nutritional therapy is often necessary during acute exacerbations of COPD, particularly in undernourished patients [7,8].

While the relationship between the underlying disease and diet is natural in conditions such as metabolic disorders, obesity, diabetes, etc., the direct effect is not so evident in chronic respiratory diseases such as COPD [9,10]. Nevertheless, a proper diet is an incredibly important component of successful COPD management. The positive impact of a diet that is appropriate in quality and quantity becomes evident even in COPD, when considering undernutrition, malnutrition, or extreme overweight, all of which significantly impair breathing [9,10]. A proper diet improves the quality of life and slows disease progression, and therefore, patient education and dietary advice are essential in this chronic respiratory disease [11]. Furthermore, even with a normal BMI, lean muscle (fat-free mass) is reduced, which negatively affects respiratory muscle function in COPD [12,13]. This is further exacerbated by systemic chronic inflammation, physical inactivity, nutritional deficiencies (such as macronutrient and/or micronutrient deficiencies and protein insufficiency), increased energy requirements due to infections, and the side effects of medication (e.g., corticosteroids) [14,15,16,17,18].

Although COPD has primarily a pulmonary manifestation, it can be associated with extrapulmonary diseases, such as cardiovascular disease, osteoporosis, metabolic syndrome, anxiety, lung cancer, and depression. These comorbidities are highly prevalent in COPD and worsen disease outcomes, impairing the quality of life of patients and impacting disease prognosis [19,20]. These observations support the pathological inflammatory response in the lungs, and it is associated with systemic inflammation throughout the body, particularly during acute exacerbations [14,21,22]. It has a serious impact on the cardiovascular system, metabolism, and skeletal muscle and can lead to the exacerbation of existing cardiovascular diseases, type 2 diabetes mellitus, hypertension, osteoporosis, deconditioning, malnutrition, muscle dysfunction, atrophy, and depression [2,14,20]. It has been observed that COPD patients who are active smokers have poorer diets than those who have quit smoking [23], and active smokers also have higher levels of oxidative stress. Oxidative stress can be reduced by dietary modification, positively influencing the levels of inflammatory markers [10]. Given the fact that 70% of COPD patients have major comorbidities, it is hypothesized that either common risk factors (aging, smoking, reduced physical activity, dietary habits leading to obesity) play a role in the co-morbidity or (less well established at present) that the combination of the individual diseases has a synergistic effect in worsening or even exacerbating the adverse effects. As a consequence, the natural course of the disease is accelerated, the functional capacity of the patient is reduced, dyspnea worsens, and in parallel, the quality of life deteriorates and the risk of mortality increases [24].

Recent epidemiological follow-up studies have reported that high intakes of antioxidant-rich foods, especially fresh fruits and vegetables, may be beneficial for respiratory function and symptoms in people with chronic respiratory disease [25,26,27]. On the contrary, high intakes of saturated fatty acids (typical in Western diets) may exacerbate airway inflammation [28,29]. Nutritional interventions play a crucial role in both the prevention and treatment of respiratory diseases and offer potential therapeutic opportunities for patients with chronic respiratory conditions. Therefore, the aim of this review was to provide an overview of these nutritional interventions in COPD and to describe their effects on respiratory function, physical activity, systemic inflammatory parameters, and the quality of life of patients.

## 2. Methods

A literature search was conducted in the PubMed, ClinicalTrials.gov, and Cochrane Central Register of Controlled Trials (CENTRAL) databases between 1 July 2018 and 1 July 2023, to collect randomized controlled trials (RCTs) and human clinical trials of nutritional interventions specifically for COPD patients. Specific and Medical Subject Headings (MESH) keywords were used in the search, including chronic obstructive pulmonary disease (COPD), exacerbation, spirometry, respiratory function, physical activity, quality of life, protein, carbohydrate, fiber, vegetables, fruits, omega-3 polyunsaturated fatty acids, probiotics, nuts, legumes, whole grains, olive oil, fish, nutritional intervention, nutritional support, dietary intervention, dietary therapy, and macronutrient supplementation, without imposing language restrictions. Between the words of the search words listed, we used the conjunction “AND” or “OR”. Indexed duplicate articles were removed, and then, the titles and abstracts were screened; those that did not meet inclusion criteria were excluded. The articles finally selected were carefully evaluated on the basis of their full text. The process of study selection is illustrated in a flow chart (Figure 1).

The aim of this state-of-the-art review was to summarize the most-recent evidence on nutritional interventions for individuals with COPD; however, for all nutritional interventions, the criteria for blindness may be compromised by caregivers or patients, even in RCT studies, and therefore, the results may be biased. In addition, there is overlap in nutritional interventions and outcomes in the studies listed below, and there is considerable heterogeneity in the respiratory function, physical condition, and quality of life of the study population. Therefore, some interventions had an effect on the quality of life of patients, physical activity, or systemic inflammatory parameters, whereas some studies had no significant results. This article is intended to be thought-provoking, and further follow-up studies should investigate the impact of long-term dietary interventions in COPD and whether these interventions play a role in the natural history, progression, and management of the disease. Therefore, we did not design a meta-analysis with a systematic literature search.

This review did not cover the intake of different vitamins, antioxidants, minerals, micronutrients, or animal experiments and in vitro studies. The aim of this state-of-the-art review was to provide an overview of the most-recent nutritional interventions in COPD according to the Population, Intervention, Comparison, and Outcomes (PICO) criteria and to describe their effects on the respiratory function, physical activity, systemic inflammatory parameters, and quality of life of patients.

### 2.1. Inclusion Criteria

-Study population: patients over 40 years of age admitted with a diagnosis of COPD.-Intervention: nutritional intervention (protein, carbohydrate, fiber, vegetables, fruits, omega-3 polyunsaturated fatty acids, probiotics, nuts, legumes, whole grains, olive oil, fish, nutritional intervention, nutritional support, dietary intervention, dietary therapy, and macronutrient supplementation).-Outcome concepts: lung function (spirometry, exacerbation), physical activity level (6-Minute Walk Test, Incremental Shuttle Walk Test), systemic inflammatory parameters (C-reactive protein, interleukins, and tumor necrosis factor alpha), quality of life (COPD Assessment Test, St George’s Respiratory Questionnaire, and EuroQol-5D), mortality risk.-Study design: randomized controlled trials and human clinical trials.-Language of publication: no language restrictions applied.-Published articles in the PubMed, ClinicalTrials.gov, and Cochrane Central Register of Controlled Trials (CENTRAL) databases.

### 2.2. Exclusion Criteria

-Animal experiments.-In vitro studies.-Vitamins, antioxidants, minerals, and micronutrients interventions.-Dietary advice without intervention.-Short-term intervention (<7 days).-Intravenous nutrition only.-Nutritional interventions for obese patients (body mass index (BMI) ≥ 30 kg/m^2^).

A total of 26 articles were included in this review, involving a total of 1811 COPD patients.

## 3. Results

Five of the studies included in this summary (Table 1) investigated specifically changes in the respiratory function of patients following nutritional interventions. Al-Azzawi MA et al. [30] reported a significant effect after supplementation with black seed oil, and their study demonstrated numerous positive effects on various aspects of COPD, particularly respiratory function, airway inflammation, and oxidative–antioxidant status. Supplementation with black seed oil significantly improved these crucial clinical parameters in COPD patients and is, therefore, specifically recommended as an adjunct therapy for COPD. Buha I et al. [31] evaluated the effectiveness of propolis and N-acetylcysteine in their study and found a significant difference in the incidence of acute exacerbations in their one-year follow-up RCT. Han MK et al. [32] investigated the safety of quercetin supplementation, which proved to be safe even at a daily dose of 2000 mg, but no significant improvements in respiratory function or total blood count parameters were measured. Lu MC et al. [33] investigated the effects of oligomeric proanthocyanidins (OPC) extracted from grape seeds on lung function and found no significant difference between the two groups—interventional group (IG) and control group (CG)—after eight weeks. However, OPC supplementation was effective in increasing antioxidant capacity and improving serum lipid levels in COPD patients. According to a Chinese retrospective analysis [34], nutritional support can effectively improve the lung function of COPD patients with respiratory failure, reduce inflammatory parameters of the blood, and also, effectively improve the therapeutic effect of patients.

With regard to systemic inflammatory parameters, changes in C-reactive protein (CRP), interleukins (ILs), and tumor necrosis factor-alpha (TNF-α) were specifically investigated as a result of nutritional interventions. Ahmadi A et al. [35] hypothesized that the daily consumption of an enriched whey drink (250 mL per day of whey drink enriched with magnesium and vitamin C) in patients with moderate to severe COPD would significantly improve quality of life, reduce levels of inflammatory parameters, and improve muscle strength. Their results showed that IL-6 concentrations were significantly reduced in the intervention group compared to the control group. Additionally, the fat-free mass index (FFMI) and handgrip strength (HGS) significantly increased in the intervention group, which led the researchers to specifically recommend this nutritional intervention. Beijers RJ et al. [36] hypothesized that the administration of resveratrol for four weeks (150 mg/d) would improve metabolic health, enhance muscle mitochondrial function, improve systemic inflammation levels, and affect body composition in COPD patients as in healthy individuals. However, the study found no significant improvement in CRP or kynurenine levels, nor did resveratrol affect the inflammatory parameters of adipose tissue.

Matheson EM et al. [37] reported on the relationship between physical activity, muscle performance, and the 6 Minute Walk Distance (6 MWD) Test, and the nutritional supplementation with a special high-protein oral nutritional supplement (ONS) significantly improved the measured clinical parameters in elderly (65 years and older), malnourished patients during their hospital stay and for 90 days after hospitalization. This improvement was observed in handgrip strength (*p* = 0.043) and nutritional status in COPD and other severe chronic diseases such as acute myocardial infarction, heart failure, and pneumonia. Their results confirmed their hypothesis about the effectiveness of this valuable nutritional intervention. Møgelberg N et al. [38] hypothesized that a high-protein diet (supplemented with protein powder) and exercise would have clinically significant effects on physical activity and peripheral muscle function in COPD patients undergoing rehabilitation. With this nutritional intervention, the intervention group significantly outperformed the control group in terms of the 6MWD at Week 12 (97 ± 93 m; *p* = 0.04). However, no significant differences were observed between the intervention and the control groups in terms of HGS, anthropometry, or dyspnea (i.e., COPD symptoms). Deutz NE et al. [39] also examined the effects of a high-protein oral nutritional supplement in malnourished, hospitalized elderly COPD patients and identified predictors of outcomes 90 days after the nutritional intervention. This intervention significantly reduced the risk of mortality (*p* = 0.03), significantly improved HGS (*p* = 0.04), and resulted in significant improvements in body weight and nutritional biomarker values within 90 days after hospital discharge. In another study, De Benedetto F et al. [40] demonstrated that dietary supplementation with coenzyme Q10 (CoQ10) and creatine administered orally for three months significantly improved functional performance (6MWT: *p* < 0.05), body composition, perception of dyspnea, well-being, and daily activities in COPD patients receiving long-term oxygen therapy (LTOT). Furthermore, positive changes were observed in the metabolic profile of patients treated, and increases in certain anti-inflammatory metabolites were also found to be significant. Aldhahir AM et al. [41] found no statistically significant difference in the Incremental Shuttle Walk Test (ISWT) distance: (IG: 342 ± 149 m; CG: 305 ± 148 m (*n* = 22); *p* = 0.1) during the six-week pulmonary rehabilitation as a result of the high-protein supplementation in the intervention group, but described that this nutritional intervention was acceptable and beneficial for patients.

The aim of the study by Karim A et al. [42] was to evaluate the effect of probiotic administration on sarcopenia and physical performance in patients with obstructive lung disease. In this randomized controlled trial, the intervention involved the use of Vivomix, a probiotic supplement, administered at a dosage of one capsule per day for 16 weeks. The probiotics reduced the plasma levels of zonulin, claudin-3, and c-terminal agrin fragment-22 (CAF22) and significantly improved handgrip strength, walking speed, and the short physical performance score (all with significant results: *p* < 0.05). The probiotic treatment also significantly reduced plasma CRP and 8-isoprostane levels, as well as markers of systemic inflammation and oxidative stress (*p* < 0.05). Overall, it was found that combined probiotic supplementation improved muscle strength and functional performance in patients with obstructive lung disease by reducing intestinal permeability and stabilizing neuromuscular junctions. 

De Brandt J et al. [43] hypothesized that 12 weeks of oral beta-alanine supplementation (3.2 g/d) would increase muscle carnosine, which acts as an abundant endogenous antioxidant and pH buffer in skeletal muscle. Their results showed that this nutritional intervention increased muscle carnosine levels by 54% without side effects. However, no beneficial changes were observed in physical performance, quadriceps function, or muscle oxidative/carbonyl stress. Nutritional interventions conducted by Ogasawara T et al. [44], Beijers RJHCG et al. [45], and Engelen M et al. [46] did not yield significant benefits in terms of physical activity, respiratory function, or quality of life between the two groups in their respective studies. On the other hand, Kerley CP et al. [47] found that the daily administration of nitrate-rich beetroot juice (BRJ; 12.9 mmol) significantly increased incremental shuttle walk test scores (ISWT) (+56 m, *p* = 0.00004). Pavitt MJ et al. found that nitrate-rich BRJ supplementation improved exercise endurance time and also improved endothelial function [48]. They described that this nutritional intervention was well tolerated and was an effective therapeutic strategy in the pulmonary rehabilitation of COPD patients without side effects [49].

In this summary, six studies examined the changes in quality of life resulting primarily from nutritional interventions. The one-year nutritional intervention program by van Beers M et al. [50] in moderate COPD significantly improved the physical activity (Δ1030 steps/day; *p* = 0.025) and quality of life of patients (EuroQol five-dimensional questionnaire; *p* = 0.009), and there was also a significant difference in terms of anxiety/depression (Δ-1.92 points; *p* = 0.037) in the intervention group compared to the control group. Additionally, this program led to improvements in weight and physical conditions, and the targeted nutritional supplementation and dietary counseling as part of this multimodal nutritional strategy proved to be cost-effective. Ingadottir AR et al. [51] also assessed the effectiveness of the nutritional intervention with a one-year follow-up in the intervention group, with a particular focus on the potential effects of snacking between meals, as well as the combined administration of snacks and oral nutritional supplements on the body weight and quality of life of COPD patients. In both groups, there was a significant increase in body weight and an improvement in the overall score of quality of life (St George’s Respiratory Questionnaire (SGRQ)) from baseline to 12 months, but the snack group scored significantly higher than the ONS group (ONS: 3.9 ± 11.0; *p* = 0.176 vs. snack: 8.9 ± 14.1; *p* = 0.041). Overall, dietary supplementation with snacks proved to be as effective as the ONS therapy in COPD patients at nutritional risk. Zhang JH et al. [52] described that nutritional and psychological intervention combined with pulmonary rehabilitation exercises could reduce the frequency of acute exacerbations of COPD and alleviate anxiety and depression symptoms, significantly improving the quality of life (SGRQ: 36.8 ± 20.8 vs. 48.6 ± 19.5; *p* <0.05).

Kim JS et al. [53] investigated the effects of omega-3 supplementation for six months in COPD patients. The significantly positive changes observed in quality of life indicated that omega-3 supplementation may have biological and clinical effects that require further investigation. Baumgartner A et al. [54] investigated the effects of personalized nutritional support to achieve calorie and protein goals in multimorbid patients with bronchopulmonary infection who were at nutritional risk due to COPD. This intervention showed a significantly positive effect on mortality risk at 30 days post-intervention (weight gain of more than 2 kg is a significant predictor of survival in COPD patients; OR: 0.47 (95% CI 0.17–1.27; *p* = 0.136) vs. 0.65 (95% CI 0.47–0.91; *p* = 0.011)). Calder PC et al. [55] showed a significant effect on patients consuming a drink containing high doses of omega-3 fatty acids, vitamin D, and high-quality protein (200 mL 2000 mg omega-3 PUFA + 10 μg vitamin D/d) as well. This targeted nutritional intervention also had positive effects on blood pressure, blood lipids, and HDL cholesterol, as well as exercise-induced fatigue and dyspnea.

## 4. Discussion

Our present study highlights several crucial dietary messages described in previous research as well [5,9,10,11,12,25,26,27,28,29]. Specifically, it emphasizes that the daily consumption of energy- and protein-rich foods, along with omega-3 supplementation, improves the nutritional status, exercise tolerance, and overall quality of life of patients. It also highlights the significance of nutrition and proper diet as modifiable risk factors in the prevention and management of COPD. In this respect, the diet of patients requires considerably more attention. This current review summarizes the latest evidence from observational and clinical studies and evaluates the outcomes of dietary interventions on the lung function, the exercise tolerance of patients, COPD progression, systemic inflammatory parameters, the quality of life of patients, and the exploration of potential underlying mechanisms. Our results show that, after different nutritional interventions, the respiratory function [30,34] and exercise capacity of patients improve [40,42,47,50] and airway inflammation [30] and systemic inflammatory parameters in the blood are significantly reduced [34,35,42], while exacerbations are reduced [31,52] and antioxidant capacity improved [30,33,42,43]. Previously, other studies have also described [56,57,58,59] that these nutritional interventions are specifically recommended in COPD because they increase respiratory muscle strength, as well as peripheral muscle strength, with concomitant improvements in metabolism and various measured clinical parameters [37,38,39,40,56,57,58,59]. In addition, an increase in the body mass index reduces the risk of mortality [39,54], improves quality of life [50,51,53,55], and reduces anxiety, depression [50,52], and dyspnea [40,55], all of which are achieved without side effects and in a cost-effective manner. Although we currently lack long-term follow-up and conclusive data comparing various diets in the context of COPD, it can already be asserted that comprehending the impact of the selected diet and emphasizing its significance pave the way for targeted future research.

The relationship between general health status, quality of life, and dietary habits is well known. There are certain physical conditions and diseases for which appropriate diets are easily accessible and have long been ingrained in public consciousness. However, there are also diseases for which the related dietary literature is less extensive, and COPD falls into this latter category. The quality of life of COPD patients can be improved with appropriate dietary interventions; disease progression can be significantly slowed, and its symptoms can be alleviated with appropriate, personalized therapy, and lifestyle changes, as confirmed by our current review. Proper nutrition is a kind of balance, which means that the body needs to consume sufficient energy, protein, vitamins, and minerals [60,61,62]. Given that severe COPD is characterized by gas exchange impairment, carbon dioxide production, i.e., carbohydrate intake, should be limited. The Respiratory Quotient (RQ) in the calorie source should be taken into account: if fats (RQ = 1) are provided, less CO_2_ is produced compared to glucose (RQ = 0.7) for the same calorie amount. The recommended nutrient ratios, with a focus on reducing carbohydrate intake to facilitate respiratory work, are as follows: 35–40% of energy from fat, 40–45% of energy from carbohydrates, and a protein intake of 1.2–1.5 g/kg of body weight (percent energy) [63]. If someone consumes less energy than is necessary, this leads to weight loss and weakened respiratory muscles, causing breathing difficulties, which leads to a loss of appetite, further weight loss, and ultimately, a vicious cycle [64,65]. Furthermore, reduced appetite is a common problem in COPD, as patients feel and complain that chewing, swallowing, and breathing require excessive effort, leading to breathlessness during meals or a sensation of bloating from swallowed air [66]. Conversely, excessive weight is also a problem in COPD (especially central, visceral obesity), as being overweight can interfere with proper lung expansion during respiration, and obesity itself can increase the oxygen demand of the body [67]. Both overweight and underweight COPD patients often experience breathlessness during eating [68], which can be alleviated by proper pre-/post-meal posture and rest [69,70].

Nutritional status plays a significant role in the progression of COPD; therefore, dietary intervention should be an essential part of treatment [71]. Patients with good nutritional status are more likely to maintain physical activity, which has an impact on their quality of life, whereas poor nutritional status reduces the chance of survival [7,71]. With an adequate diet and regular exercise, the risk of death is significantly reduced and the physical fitness and quality of life of patients can be significantly improved, which also influences the prognosis of the disease [7]. Chronic obstructive pulmonary disease is characterized by a mixed form of malnutrition, by a decrease in visceral and muscle proteins, the depletion of fat stores, weakened immune defense, and ultimately, energy [12]. Weight loss is primarily a consequence of skeletal muscle atrophy [72]. In COPD, muscle mass loss (sarcopenia) is very similar to other conditions associated with chronic cachexia, such as heart failure, renal failure, and sarcopenia in cancer [73]. In these chronic diseases, physical wasting is associated with both reduced survival and poorer functional and quality of life outcomes, which also justifies the timely initiation of nutritional therapy and aligns with the effectiveness of the aforementioned studies, improving the quality of life of patients, increasing their physical activity and well-being [74,75]. A common feature of these chronic diseases is the presence of elevated blood levels of pro-inflammatory mediators (e.g., TNF-α, IL-6, INF-γ) [76,77], whereas levels of anabolic hormones (e.g., testosterone, insulin-like growth factor) are lower than normal [78]. The exact underlying cause of muscle atrophy is not precisely known; however, some studies suggest that TNF-α, while activating nuclear factor kappa-B (NF-kB) and inducting nitric oxide synthase (NOS), triggers myosin degradation and apoptosis of muscle cells [7,79].

Although COPD is a severe and chronic illness, patients can take several measures to facilitate their breathing. Although the disease is not curable, certain lifestyle changes (such as quitting smoking, increasing physical activity, adopting an optimal quality and quantity of nutrition and dietary supplementation) are necessary for its management [80]. The goals of therapy are to reduce symptoms, slow down disease progression, improve overall health, increase physical activity, prevent and treat exacerbations, and ultimately, improve quality of life [81]. Diet is described as a modifiable risk factor for the development and progression of chronic diseases [82], and emerging evidence increasingly highlights its preventive role in obstructive respiratory lung diseases as well [83,84]. Furthermore, dietary factors can modify the effects of harmful environmental factors and/or the genetic predisposition to lung diseases [85]. Increasing body weight improves the performance of respiratory muscles, increases the exercise capacity and ventilation of patients, and reduces the risk of acute exacerbations, all of which are supported by our current review. In undernourished patients, early nutritional therapy with protein-rich and high-calorie foods and, if necessary, formula supplementation is warranted [84].

A high antioxidant content has a positive effect on the quality of life of patients with COPD, with the Mediterranean diet known to be extremely rich in vegetables and fruits—i.e., antioxidants—and low in saturated fats [86,87]. Oxidative stress plays an important role in the development of age-related chronic diseases, and there is evidence that poor nutrition can increase the level of oxidative stress, the risk of systemic inflammation and chronic diseases, as well as tissue damage, respiratory inflammation, COPD exacerbation, and abnormal immune responses [88]. Recent follow-up epidemiological studies have reported potential beneficial effects of antioxidant-rich foods, especially fresh fruits and vegetables, on respiratory function and symptoms in individuals with chronic respiratory diseases [89,90,91]. A randomized controlled trial reported that COPD patients following a diet rich in fruits and vegetables had a significant annual increase in forced expiratory volume in 1 s (FEV_1_) compared to controls (*p* = 0.03) [27]. Together, these observations suggest that fruit and vegetable consumption is an important determinant of respiratory function and COPD risk [89].

Research has described that antioxidants and flavonoids in plant-based foods reduce respiratory inflammation, resulting in improvements in FEV_1_ and forced vital capacity (FVC) in patients with chronic respiratory diseases [84,92,93]. Increasing fiber intake is also a potential way to treat respiratory symptoms due to its anti-inflammatory effects [94]. The Mediterranean diet has long been considered the healthiest diet [95,96], with health benefits attributed largely to its fiber content, antioxidants, proteins, and moderate fat intake, primarily monounsaturated fatty acids and omega-3 polyunsaturated fatty acids (PUFAs) [95]. Omega-3 PUFAs have received significant attention for their anti-inflammatory properties and anticoagulant effects, thereby reducing the risk of cardiovascular diseases [93]. They are nutritionally essential and can be obtained mainly from seafood (e.g., fatty fish) [93]. On the other hand, omega-6 fatty acids, including linoleic acid and its long-chain derivative, arachidonic acid, which are found primarily in vegetable oils (such as soybean, corn, and sunflower oil), dairy products, and eggs, have been described to have inflammatory effects [97]. It is hypothesized that the Western diet with increased consumption of omega-6 fatty acids and decreased consumption of omega-3 fatty acids has contributed to the rising prevalence of chronic inflammatory diseases [98,99]. Some previous studies suggest that increased consumption of fish and plant-based sources of omega-3 PUFAs may reduce the severity of COPD (a claim supported by our current research), indicating that healthy nutrition can be a beneficial intervention for COPD patients [84,100,101].

Observational studies have reported independent beneficial effects of whole grain intake on respiratory function [83,84,102] and on symptoms of chronic respiratory diseases [50]. Whole grains are rich in phenolic acids, flavonoids, phytic acid, vitamin E, selenium, and essential fatty acids, which can contribute additively to their beneficial effects [103]. Due to their high fiber content, they also have antioxidant and anti-inflammatory properties, and increased fiber intake has been associated with lower serum levels of CRP, IL-6, and TNF-α and higher adiponectin [104,105]. When comparing different types of fiber (such as cereal, fruit, and vegetable fiber), researchers have observed the most-significant beneficial correlation with cereal fiber intake, especially among active smokers and those who have quit smoking, but there is also scientific evidence supporting the beneficial effects of increased fruit and vegetable fiber intake [84,106]. The results of the ECLIPSE study on COPD patients showed that increased consumption of “healthy” foods such as fruits, fish, tea, dairy products, whole grains, etc., was associated with improved respiratory function, better prognosis, quality of life, and exercise tolerance, as well as lower inflammatory parameters (CRP, white blood cells, surfactant protein D, etc.) [107].

Finally, with the widespread consumption of coffee, there is growing interest in the potential role of caffeine in respiratory health. Meta-analyses have shown a correlation between regular—non-decaffeinated—coffee consumption and improvements in respiratory function, as well as reductions in mortality from respiratory diseases, but not COPD [108]. This may be attributed to the bronchodilatory, anti-inflammatory effects of caffeine, as well as the antioxidant and anti-inflammatory effects of its polyphenols. However, smoking is a significant confounding factor in these studies, as it can accelerate the metabolism and clearance of caffeine and it can also blunt the beneficial effects of caffeine due to its potent inflammatory effects [108].

### 4.1. Potentially Harmful Foods for Respiratory Function

Health promotion programs should provide specific advice to reduce the consumption of red and processed meats and meat products such as pickled, salted, and smoked meats, bacon, and processed and semi-prepared industrial products (e.g., sausages), as the available literature suggests a negative correlation between frequent consumption of these foods and respiratory function in COPD [109]. In addition, preservatives such as nitrites are added to processed meats during production, which create reactive peroxynitrite, which can further exacerbate the inflammatory cascade in the airways, causing further cellular dysfunction and inhibiting mitochondrial respiration. In animal studies, chronic nitrite exposure has caused emphysema-like pathological changes in the lungs [110]. Smoking and increased consumption of high-sodium cured meats may also increase bronchial hyperreactivity and additively trigger inflammation. Furthermore, meats contain high levels of saturated fatty acids, which can further exacerbate inflammation in the airways, which is associated with impaired respiratory function [111], as well as an increased risk of coronary and metabolic diseases. In contrast, low-fat dairy products have a protective effect, probably due to their anti-inflammatory properties [112,113]. These results clearly emphasize the very important public health significance of interventions targeting modern, unhealthy diets.

### 4.2. Limitations

The present review is not a systematic one and may have omitted research on dietary factors affecting the quality of life, physical activity, and inflammatory parameters of COPD patients. Due to the scope limitations of the final publication, only the most-recent RCT studies were included in this review, i.e., published articles from the last five years on this very important dietetic topic, which may be a biasing factor. We focused primarily on RCT studies in English, so other publications in foreign languages may have been omitted from this summary study, and conference abstracts and meta-analysis were not included. This review evaluated the evidence on dietary modification, and it appears this modification may be advantageous in patients with chronic respiratory disease and may also be beneficial in COPD; however, further studies are needed to accurately assess and describe the effects of dietary interventions in long-term follow-up studies. It is extremely difficult to design such studies because of the confounding factors of comorbidities, medications, possible obesity or malnutrition, and environmental exposure, but while such studies are being designed, it seems appropriate to consider formulating dietary recommendations and to consider their role in individuals at risk. On the other hand, an additional bias is that dietary influences, as well as the Mediterranean diet affects other organs and organ systems (especially the cardiovascular system), which clearly introduces a confounding bias in the results of the different studies, since measuring respiratory function alone without taking cardiac function into account is very complex, and the RCTs presented did not perform this in any of the cases. The degree of the COPD severity, physical condition, nutritional status, body composition, age, sex, and co-morbidities of the patients studied varied widely across the different studies, and selection bias could have occurred. The studies are, therefore, not comparable, and the description of the research programs is very different.

## 5. Conclusions

In the context of the review presented, although the research studies involved a small number of patients and the researchers were generally only able to conduct follow-ups for a few weeks, their observations support the association between a proper quality diet and COPD, which has previously been confirmed by a number of reputable publications. Unfortunately, to date, studies with appropriate strength and long-term follow-up in clinical settings are lacking, making it extremely challenging to predict the future implementation of these highly important and pioneering studies. Nonetheless, COPD is of paramount importance in the field of chronic diseases; its prevalence is widespread; mortality is high; decades of treatment are extremely expensive; no form of pharmacotherapy can halt the progressive decline in respiratory function. Achieving and maintaining an appropriate nutritional status is, therefore, becoming increasingly important in the care of chronic pulmonary patients, alongside dietary modification and the associated improvements in quality of life. In fact, people with chronic obstructive pulmonary disease typically exhibit a loss of appetite and experience compromised respiration, high and significant medication intake and weight loss, potentially leading to further deterioration in their condition. Furthermore, the increased energy expenditure associated with the disease can further exacerbate weight loss and deterioration in respiratory function, thus potentially leading to a vicious cycle. The aim of a good diet is to achieve and maintain adequate nutritional status, stop weight loss, and regain energy. It is recommended to eat foods that are rich in energy and protein and omega-3 fatty acids and that do not bloat. Evidence from observational and intervention studies shows that the Mediterranean-style diet (e.g., increased intake of protein, omega-3 PUFA) is one of the healthiest diets that can protect against major chronic diseases. These observations have been confirmed by the present review and are important for both the prevention and treatment of these diseases. Overall, although further confirmation is needed, studies consistently suggest that a high-quality, whole-food diet plays a significant role in the quality of life and mortality of patients. Additionally, a healthy diet can also counteract visceral fat deposition, associated systemic inflammation and oxidative stress, mitochondrial dysfunction, as well as insulin resistance, thus potentially offering an opportunity to address the risk of metabolic disorders observed in some COPD patients (obesity and/or abdominal visceral obesity). Moreover, dietary factors can induce modifications in the gut microbiome [114], which can have a positive impact on the immune system, systemic inflammation, metabolism, and cardiometabolic health. In conclusion, on the basis of the evidence presented in this overview and in line with the available interventional studies, our recommendation to promote a healthy lifestyle in COPD includes increasing physical activity alongside a balanced diet, complete smoking cessation, and consistent application of pharmacological therapy. This comprehensive intervention offers the potential for targeted, early, and effective therapy.

## Figures and Tables

**Figure 1 nutrients-15-03902-f001:**
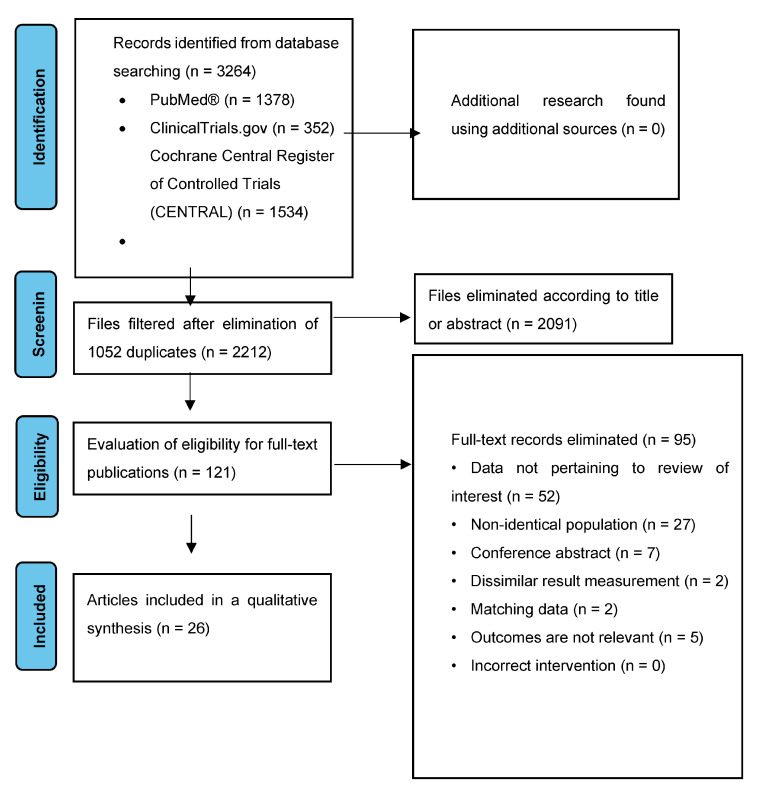
Flow diagram of articles involved in the selection process.

**Table 1 nutrients-15-03902-t001:** Nutrition intervention studies in COPD patients.

Study	Design	Mean Follow-Up	Country	Sample Size	Average Age (Year)	Sex (Male/Female)	Intervention	Main Results
Al-Azzawi MA et al. [30]	RCT	3 months	Egypt	91	55.2 ± 4.3	69%/31%	Treated with 1 g of 100% pure cold-pressed black seed oil twice daily.	Significant reduction in oxidant and inflammatory markers. A significant improvement in pulmonary function tests versus baseline levels and control group (CG).
Buha I et al. [31]	RCT	1 year	Serbia	46	65 ± 8	63%/37%	One patient (AS-600) received 600 mg of NAC + 80 mg of propolis, while the other (AS-1200) received 1200 mg of NAC + 160 mg of propolis.	Compared to placebo, AECOPD frequency was significantly lower only in AS-1200 (*p* = 0.009). Compared to placebo, the relative risk for exacerbation was 0.29 in AS-600 and 0.13 in AS-1200.
Han MK et al. [32]	RCT	7 days	Michigan	9	68 ± 6	56%/64%	Quercetin at 500, 1000 or 2000 mg/d.	Quercetin was safely tolerated up to 2000 mg/d based on lung function, blood profile, and COPD assessment test questionnaire.
Lu MC et al. [33]	RCT	8 Weeks	Taiwan	27	71 ± 2	-	150 mg/d oligomeric proanthocyanidins extracted from grape seed suppl.	Oligomeric proanthocyanidins supplementation significantly reduced the concentration of malondialdehyde and superoxide dismutase.
Wang L et al. [34]	RCT	Retrospective analysis	China	127	70 ± 3	53%/47%	Enteral and parenteral nutrition support.	Lung function improved, and inflammatory factor levels decreased (*p* < 0.05). The levels of serum albumin, prealbumin, serum hemoglobin, and serum transferrin increased after nutritional support (*p* < 0.05).
Ahmadi A et al. [35]	RCT	8 weeks	Iran	44	62 ± 7	100% male	Intervention group (IG) daily received 250 mL of whey beverage fortified with magnesium and vitamin C.	This nutritional intervention decreased inflammatory cytokine levels, improved indices of skeletal muscle mass and muscle strength, and ultimately, increased quality of life (QOL).
Beijers RJ et al. [36]	RCT	4 weeks	Netherlands	21	67 ± 9	57%/43%	Resveratrol supplementation (150 mg/nap).	They did not confirm previously reported positive effects of resveratrol on skeletal muscle mitochondrial function in patients with COPD, but showed an unexpected decline in lean mass.
Matheson EM et al. [37]	RCT	90 days	USA	354 (124 COPD)	≥65	50.8%/49.2%	Received a high dose of protein and beta-hydroxy-beta-methylbutyrate containing oral nutritional supplement (ONS).	ONS provided during hospitalization and up to 90 days post-discharge improves handgrip strength (HGS) in malnourished older adults.
Møgelberg N et al. [38]	RCT	12 weeks	Denmark	10	68 ± 12	30%/70%	High-protein diet.	High-protein diet combined with physical exercise had a clinically relevant effect on walking distance: 6MWD (97 ± 93 m, *p* = 0.04).
Deutz NE et al. [39]	RCT	90 days	USA	214	74.5 ± 7.3	47.2%/52.8%	High-protein oral nutritional supplement (ONS) containing β-hydroxy-β-methylbutyrate (HMB).	Improved handgrip strength, body weight, and nutritional biomarkers within a 90-day period after hospital discharge.
De Benedetto F et al. [40]	RCT	2 months	Italy	90	73 ± 7	75.5%/24.5%	Received 160 mg Coenzyme QTer^®^ + 170 mg creatine.	Supplemented patients showed improvements in 6MWT (51 ± 69 versus 15 ± 91 m, *p* < 0.05), body cell mass and phase angle, sodium/potassium ratio, dyspnea indices, and ADL score.
Aldhahir AM et al. [41]	RCT	6 weeks	U.K.	68	70 ± 9	62%/38%	High-protein supplementation during pulmonary rehabilitation.	No significant difference in Incremental Shuttle Walk Test (ISWT) distance: (IG: 342 ± 149 m; CG: 305 ± 148 m; *p* = 0.1).
Karim A et al. [42]	RCT	16 weeks	United Arab Emirates	104	66.9 ± 3.4	100% male	Vivomix 112 billion *, one capsule a day.	Probiotics reduced plasma zonulin, claudin-3, and CAF22, along with an improvement in HGS, gait speed, and Short Physical Performance Battery (SPPB) scores (all *p* < 0.05).
De Brandt J et al. [43]	RCT	12 weeks	Belgium	40	65 ± 6	70%/30%	Beta-alanine supplementation (3.2 g/d).	Beta-alanine supplementation is efficacious in augmenting muscle carnosine (+54% from mean baseline value) without side effects.
Ogasawara T et al. [44]	RCT	2 weeks	Japan	45	77 ± 9	91%/9%	Received 1 g/d of eicosapentaenoic acid-enriched (EPA) oral nutrition supplementation.	EPA-enriched ONS supplementation had no significant benefit on lean body mass (LBM) and muscle mass.
Beijers RJHCG et al. [45]	RCT	7 days	Netherlands	18	66.6 ± 7.5	72.2%/27.8%	Sodium nitrate (beetroot juice) ingestion (∼8 mmol/d).	7 days of sodium nitrate supplementation does not modulate mechanical efficiency and blood pressure in COPD.
Engelen M et al. [46]	RCT	1 month	USA	32	-	-	3.5 g of EPA + DHA/2.0 g of EPA + DHA or placebo capsules/	Daily omega-3 (EPA + DHA) supplementation induces a shift toward a positive daily protein homeostasis. Extremity lean mass increased.
Kerley CP et al. [47]	RCT	2 weeks	Ireland	8	-	-	Daily nitrate-rich beetroot juice (BRJ; 12.9 mmol).	BRJ supplementation was associated with significantly increased NOx (*p* < 0.05) and a 14.6% increase in ISWT distance (+56 m, *p* = 0.00004).
Pavitt MJ et al. [48]	RCT	4 weeks	U.K.	20	-	-	140 mL of nitrate-rich beetroot juice (12.9 mmol nitrate) (BRJ).	Nitrate-rich BRJ supplementation prolonged exercise endurance time in the IG as compared with the CG: 194.6 (147.5–411.7) s vs. 159.1 (121.9–298.5) s.
Pavitt MJ et al. [49]	RCT	8 weeks	U.K.	122	70 ± 8	56%/44%	140 mL of nitrate-rich beetroot juice (12.9 mmol nitrate).	Change in ISWT distance +60 m (10, 85) vs. +30 m (0, 70), *p* = 0.027, and estimated treatment effect on systolic blood pressure -7 mmHg.
van Beers M et al. [50]	RCT	12 months	Netherlands	81	62.5 ± 0.9	51%/49%	3 portions of nutritional supplementation per day (enriched with leucine, vitamin D, and polyunsaturated fatty acids).	Physical activity was higher in nutrition than in placebo (Δ1030 steps/day, *p* = 0.025); weight gain in nutrition (Δ1.54 kg, *p* = 0.041); improved EQ-5D (*p* = 0.009).
Ingadottir AR et al. [51]	RCT	12 months	Iceland	34	72 ± 8	29%/71%	Hospitalized patients were randomized to ONS (*n* = 19) or snacks (*n* = 15) providing 600 kcal and 22 g of protein a day.	The SGRQ-C TS improved from baseline to 12 months in both groups (score of 3.9 ± 11.0 (*p* = 0.176) in the ONS group and score of 8.9 ± 14.1 (*p* = 0.041) in the snacks group).
Zhang JH et al. [52]	RCT	12 months	China	260	65 ± 10.4	86%/14%	The IG was given nutritional support and complex pulmonary rehabilitation with psychological intervention.	The number of acute exacerbations was significantly reduced. PaO_2_ was significantly higher than in the control group. The anxiety score (4.1 ± 2.2) vs. (5.6 ± 2.7), depression score (4.1 ± 2.0) vs. (5.5 ± 2.6). and St George’s Score (36.8 ± 20.8) vs. (48.6 ± 19.5) were significantly decreased.
Kim JS et al. [53]	RCT	6 months	Columbia	40	67.5 ± 6.5	55%/45%	Daily administration of high-dose fish oil capsules for six months.	Quality of life (SGRQ) improved significantly in COPD (4-point improvement in the SGRQ; *p* = 0.01).
BaumgartnerA et al. [54]	RCT	30 days	Switzerland	378 (91 COPD)	73.5 ± 13.5	55.1%/44.9%	Individualized nutritional support to reach protein and energy goals.	Individualized nutritional support to reach calorie and protein goals showed beneficial effect on mortality risk in the subgroup of patients with respiratory tract infection.
Calder PC et al. [55]	RCT	12 weeks	Norway	45	69.5	51%/49%	200 mL of targeted medical nutrition: 2 g omega-3 PUFA + 10 μg vitamin D3/d.	Reductions in exercise-induced fatigue (*p* = 0.0223), dyspnea (*p* = 0.0382), and systolic blood pressure (*p* = 0.0418) were observed.

RCT: randomized controlled trial; ONS: oral nutritional supplement; HGS: handgrip strength; CHO: carbohydrates; LEU: leucine; EQ-5D: EuroQol-5 Dimension; SGRQ: George’s Respiratory Questionnaire; USA: United States of America; U.K.: United Kingdom; FFM: fat-free mass; COPD: chronic obstructive pulmonary disease; EPA: eicosapentaenoic acid; 6MWD: 6 Minute Walk Distance; PUFA: polyunsaturated fatty acid; SPPB: Short Physical Performance Battery; CAF22: c-terminal agrin fragment-22; DHA: docosahexaenoic acid; IG: intervention group; CG: control group; QoL: quality of life; BRJ: beetroot juice; ISWT: Incremental Shuttle Walk Test; NOx: plasma nitrate/nitrite; eNO: exhaled nitric oxide; LBM: lean body mass; OPC: oligomeric proanthocyanidin; * Each capsule contains 112 billion live bacteria (*Streptococcus thermophilus* DSM 24731, bifidobacteria (*B. longum* DSM 24736, *B. breve* DSM 24732, DSM 24737), lactobacilli (DSM 24735, DSM 24730, DSM 24733, *L. delbrueckii* subsp. *bulgaricus* DSM 24734) along with maltose, anti-caking agent: silicon dioxide.

## Data Availability

Data sharing is not applicable to this article as no new data were created nor analyzed in this study.

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
