# Peer review of "The Possible Role of Food and Diet in the Quality of Life in Patients with COPD—A State-of-the-Art Review"

_nutrients, 2023, doi:10.3390/nu15183902_

Round 1

Reviewer 1 Report (Previous Reviewer 2)

With the modifications made by the authors, it is fine now.

Author Response

Thank You very much!

Sincerely: János T Varga

Reviewer 2 Report (New Reviewer)

This paper presents a commendable work of systematic review of RCT published in the last 5 years which have had the objective of evaluating the efficacy of numerous and heterogeneous interventions on the nutrition of patients with COPD.The authors found 26 small studies (9-260 COPD subjects) of short duration (mostly a few weeks) evaluating the efficacy of many heterogeneous dietary interventions against  many and heterogeneous clinical, functional, and biological outcomes.From what is written in the abstract it is assumed that the conclusions are consequent to the results of the studies found in the systematic review. Unfortunately, in the discussion paragraph there is a long description of topics present in the literature prior to the systematic review and only a fleeting attention to the results of the same review and only in order to discuss its limitations.I suggest that the authors rewrite the discussion and conclusion paragraphs to adequately value the commendable literature review work performed, which confirmed that there are many authoritative and plausible opinions on the diet/COPD relationship, sometimes supported by observational data. Unfortunately, experimental studies of adequate power are absent and it is very difficult to hypothesize their realization in the future.

Author Response

We are sincerely grateful for the valuable critiques provided.

  • We have rephrased the conclusion and highlighted the requested modifications in red within the manuscript.
  • In the discussion section, we have also indicated in red a more precise evaluation of the results from this up-to-date literature review.
  • Our manuscript was proofread by a native English speaker.
  • Our references have been reduced, so it is now below 15%.

My utmost gratitude to the Reviewers for their valuable professional insights, my co-authors and I believe that our research's scientific value has significantly improved.

Sincerely: János Tamás Varga

Round 2

Reviewer 2 Report (New Reviewer)

No more comments

This manuscript is a resubmission of an earlier submission. The following is a list of the peer review reports and author responses from that submission.

Round 1

Reviewer 1 Report

The manuscript entitled “The possible role of food and diet in the quality of life of 2 COPD patients - A review by Mónika Fekete et al., focuses on nutritional interventions in COPD and their effects on respiratory function, physical activity, systemic inflammatory parameters, and quality of life of patients. I have an important comment:

-A literature search has been conducted in the PubMed and ClinicalTrials.gov databases to collect randomized controlled trials. It is needed, mandatory to include another database such as  Cochrane Central Register of Controlled Trials (CENTRAL).

Dear Editor,

The manuscript entitled “The possible role of food and diet in the quality of life of 2 COPD patients - A review” by Mónika Fekete et al., falls within journal aim and scope. However, in my opinion, it could be accepted for publication only after major changes.

Author Response

Thank You for the Reviewer's time and valuable comments, the RCTs published in the Cochrane Central Register of Controlled Trials (CENTRAL) database on this topic have been reviewed and incorporated into the manuscript.

Reviewer 2 Report

This is a systematic review of dietary factors affecting patients with COPD. It is well written but includes RCTs with very small sample sizes (8, 9 or 10 patients) which is a bias in itself. The authors should detail this bias in a limitations section.

On the other hand, they should also explain in the limitations that the Mediterranean diet also has effects on other organs and systems (especially cardiovascular) which generates a clear confounding bias in their results: measuring respiratory function without taking cardiac function into account is very complex and the RCTs have not done so. 

The authors should clarify these terms.

Author Response

We appreciate the Reviewer's time and valuable comments, we have included these thoughts in the manuscript in a limitation, marked in red. The present review is not a systematic one and may have omitted research on dietary factors affecting quality of life, physical activity and inflammatory parameters in COPD patients. Due to the scope limitations of the final publication, only the most recent RCT studies have been included in this review, i.e. published articles from the last five years on this very important dietetic topic, which may be a biasing factor. We focused primarily on RCT studies in English, so other publications in foreign languages may have been omitted from this summary study, and conference abstracts and meta-analyses are not included in this summary. This review evaluates the evidence on dietary modification and it appears this modification may be advantageous in patients with chronic respiratory disease and may also be beneficial in COPD, but further studies are needed to accurately assess and describe the effects of dietary interventions in long-term follow-up studies. It is extremely difficult to design such studies because of the confounding factors of comorbidities, medications, possible obesity or malnutrition, and environmental exposure, but while such studies are being designed, it seems appropriate to consider formulating dietary recommendations and to consider their role in individuals at risk. On the other hand, an additional bias is that dietary influences as well as the Mediterranean diet affect other organs and organ systems (especially the cardiovascular system), which clearly introduces a confounding bias in the results of the different studies, since measuring respiratory function alone without taking cardiac function into account is very complex, and the RCTs presented did not do this in any of the cases. The degree of COPD severity, physical condition, nutritional status, body composition, age, sex, co-morbidities of the patients studied varied widely across the different studies, and selection bias could have occurred. The studies are therefore not comparable and the description of the research programmes is very different.
